# Learning predictable and robust neural representations by straightening image sequences

**Xueyan Niu**[1]    **Cristina Savin**[1,2]    **Eero P. Simoncelli**[1,3]

[1]Center for Neural Science, New York University
[2]Center for Data Science, New York University
[3]Center for Computational Neuroscience, Flatiron Institute

`{xn314, csavin, eero.simoncelli}@nyu.edu`

## Abstract

Prediction is a fundamental capability of all living organisms, and has been proposed as an objective for learning sensory representations. Recent work demonstrates that in primate visual systems, prediction is facilitated by neural representations that follow straighter temporal trajectories than their initial photoreceptor encoding, which allows for prediction by linear extrapolation. Inspired by these experimental findings, we develop a self-supervised learning (SSL) objective that explicitly quantifies and promotes straightening. We demonstrate the power of this objective in training deep feedforward neural networks on smoothly-rendered synthetic image sequences that mimic commonly-occurring properties of natural videos. The learned model contains neural embeddings that are predictive, but also factorize the geometric, photometric, and semantic attributes of objects. The representations also prove more robust to noise and adversarial attacks compared to previous SSL methods that optimize for invariance to random augmentations. Moreover, these beneficial properties can be transferred to other training procedures by using the straightening objective as a regularizer, suggesting a broader utility of straightening as a principle for robust unsupervised learning.

## 1   Introduction

All organisms make predictions, and their survival generally depends on the accuracy of these predictions. In simple organisms, these predictions are reflexive and operate over short time scales (e.g., moving toward or away from light, heat, or food sources). In more complex organisms, they involve internal state (memories, plans, emotions) and operate over much longer timescales. Prediction has the potential to provide an organizing principle for overall brain function, and a source of inspiration for learning representations in artificial systems. However, natural visual scenes evolve according to highly nonlinear dynamics that make prediction difficult. Recent experiments – both perceptual (in humans) and neurophysiological (in macaques) – indicate that the visual system transforms these complex pixel dynamics into *straighter* temporal trajectories [22, 23]. As an alternative to full temporal prediction, straight representations facilitate predictability through linear extrapolations (Fig. 1A, red arrows). Yet, the utility of straightening as a learning principle for organizing representations remains underexplored.

Here, we ask whether straightening is sufficiently powerful to be used as a primary objective in self-supervised learning (SSL). Our contributions are as follows:

- We developed an SSL objective function that aims to straighten spatio-temporal visual inputs. We demonstrated on simulated data that this objective, coupled with a whitening regularizer

38th Conference on Neural Information Processing Systems (NeurIPS 2024).

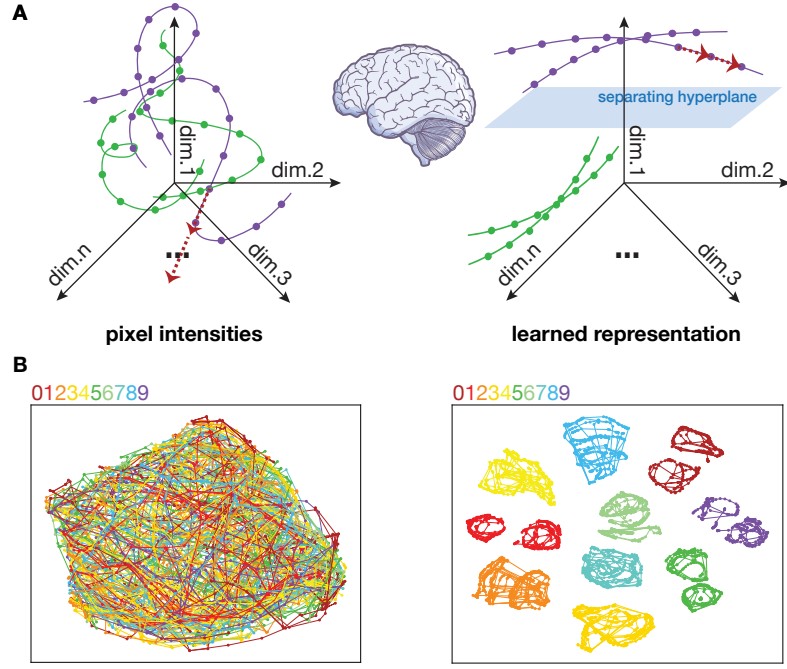

**Figure 1:** Learning straightened representations. **A**. Illustration of temporal trajectories of four translating digit sequences, in the space of pixel intensities (left), and in a straightened representation (right). Color indicates digit identity. **B**. The actual two-dimensional t-SNE rendering of 20 temporal trajectories for each of the ten translating digits from our model. Initial (pixel intensity) representation is highly curved and entangled (left). Although the straightening objective is unsupervised (no object labels), the learned representation clearly isolates the trajectories corresponding to different digits (right).

that prevents representational collapse, can successfully straighten visual inputs containing both geometric and photometric transformations.

- We show that the trained network is effective in extracting and predicting various visual attributes including object identity, location, size, and orientations.
- We provide geometric intuition for how straightening yields class separability.
- We show that representations learned by straightening are significantly more robust than those learned by multi-view invariance, when trained on the same model architecture and dataset. Moreover, straightening can be used as a regularizer to enhance the robustness of state-of-the-art SSL methods.

The implementation can be found at https://github.com/xyniu1/learning-by-straightening.

## 1.1 Related work

**Temporal invariance.** Many successful SSL methods aim to learn representations that are invariant to simple transformations, with additional regularization incorporated to prevent representational collapse (e.g. constant response, independent of input). Depending on the exact implementation of these components, three categories of SSL objectives are identified in a recently published review [2]: 1) contrastive SSL [7, 11], which encourages representation similarity between two augmented views of the same image (positive pairs) and dissimilarity of different images (negative pairs); 2) self-distillation [18, 8, 6], which uses two different encoders to process two views of the same image, and maps the two representations by a predictive projection; and 3) canonical correlation analysis [3, 35, 14] which aims to whiten the cross-correlation matrix of neural representations estimated over augmented pairs of the same image. Most of the invariance-based methods subserve tasks that operate on static images rather than spatio-temporal inputs. This is because invariance is a strict constraint and equating representations over time necessarily means eliminating the time-varying features in

the inputs. For example, for a video that contains moving objects, learning invariant representations across frames may help to encode the identity of the object but not its location or relative size. In contrast, straightening is designed not only to capture all predictable features in the spatio-temporal inputs (static and dynamic alike) but also predict their future states using a predefined operator.

**Temporal prediction.** Temporal prediction as a fundamental goal for learning visual representations dates back to [16]. Many theories rooted in predictability have been successful in characterizing the properties of early visual areas [4, 28, 15]. Invariance is the simplest form of prediction. Linear predictions have been extensively studied. Notably, [27, 21] achieved great success in unsupervised object recognition by learning a linear predictor that maps current states to future states for each step into the future. In [25], this paradigm was extended to allow a context-dependent, dynamic selection of linear predictors. Straightening differs from these methods in that it is parameter-free and its predictions can adapt to different contexts, while the previous methods rely on parametrization that scales quadratically with the feature dimension. The work most similar to ours is [17], which uses auxiliary architectural elements such as phase-pooling and further relies on an autoencoder structure and a pixel-level prediction loss to prevent information collapse. Our solution uses a much simpler architecture, and we also provide a more extensive quantitative evaluation of the resulting straightened representations.

**Straightening and robustness.** Although straightening has been documented in human perception and macaque physiology, it is not an inherent property of deep neural networks [22], including supervised and self-supervised recognition networks, and video prediction networks [31, 19]. Some non-parametric formulations of early visual processes demonstrate a degree of straightness, but the effect does not seem to propagate to downstream layers [22]. Recently, [31, 19] demonstrated that straightening can be an emerging property of robustified networks: if networks are trained to tolerate Gaussian noise or adversarial perturbations, they can generate straightened responses without being explicitly trained to do so. In this work, we provide the complementary observation: if networks are trained to straighten, the representation is robust to corruptions including Gaussian noise and adversarial perturbations.

Apart from straightening, other learning objectives that exploit the temporal structures of natural video statistics have also been shown to improve adversarial robustness, such as temporal classification (classifying frames to the episode they belong to) and temporal contrastive learning (temporally adjacent frames are used as positive examples) [26, 29].

## 2 Straightening videos

**Objective function.** We aim to learn a representation of video frames that follows a straighter trajectory over time by transforming each frame, $\mathbf{x}_t$, into network response $\mathbf{z}_t = f_\theta(\mathbf{x}_t)$, where $f_\theta$ denotes the learned transformation, parameterized by vector $\theta$. We measure straightness of an output sequence $\{\mathbf{z}_t\}_{t=1}^T$ as the average cosine similarity (normalized dot product) between the two successive difference vectors of any three temporally adjacent points. Our goal is to optimize parameters $\theta$ to maximize straightness, or minimize the loss:

$$\mathcal{L}_{\text{straightness}} = -\mathbb{E}\left[\frac{\langle \mathbf{z}_{t+1} - \mathbf{z}_t, \mathbf{z}_{t+2} - \mathbf{z}_{t+1} \rangle}{\|\mathbf{z}_{t+1} - \mathbf{z}_t\|\|\mathbf{z}_{t+2} - \mathbf{z}_{t+1}\|}\right]. \tag{1}$$

where the expectation is taken over sequences and time, $t$. This objective is invariant to rescaling of responses and is bounded within $[-1, 1]$. By default, the straightness loss is applied to the output layer, but it can be applied to any (or several) layer(s) of a network.[1] Once straightness is established, one-step prediction takes the form of linear extrapolation: $\mathbf{z}_{t+2} = 2\mathbf{z}_{t+1} - \mathbf{z}_t$.

Straightness alone is not sufficient to learn meaningful representations because it can be minimized by trivial solutions ($\mathbf{z}_t = ct$ or $\mathbf{z}_t = c$ for $\forall t$). To avoid this form of collapse, we incorporate a form of regularization borrowed from [3], which essentially aims to statistically whiten the outputs using two terms. First, a variance term for each output dimension, essentially preventing different inputs from collapsing to the same output, $\mathcal{L}_{\text{variance}} = \mathbb{E}\left[\frac{1}{d}\sum_{i=1}^d \max\left(0, 1 - S\left(z_t^i, \epsilon\right)\right)\right]$, where $S(x, \epsilon) = \sqrt{\text{Var}(x) + \epsilon}$, $d$ is the output dimensionality, and $\epsilon = 10^{-4}$. Second, a covariance term decorrelates

---

[1]For intermediate layers, straightness can be computed on the embeddings vectorized over space and channels.

each pair of output dimensions to minimize redundancies, $\mathcal{L}_{\text{covariance}} = \mathbb{E}\left[\frac{1}{d}\sum_{i\neq j}[\text{Cov}(\mathbf{z}_t)]^2_{i,j}\right]$. Taken together, the complete learning objective to minimize is given by

$$\mathcal{L} = \mathcal{L}_{\text{straightness}} + \alpha\mathcal{L}_{\text{variance}} + \beta\mathcal{L}_{\text{covariance}}, \tag{2}$$

where hyperparameters $\alpha$ and $\beta$ control the relative strength of the two regularizers.

**Comparing straightening with invariance.** To assess the benefits of straightening, we compare it to the more common invariance objective, which encourages outputs of distinct views of the same scene to be similar to each other:

$$L_{\text{invariance}} = \mathbb{E}\left[\frac{1}{d}\|\mathbf{z}_t - \mathbf{z}_{t_0}\|^2_F\right], \tag{3}$$

where $F$ denotes the Frobenius norm, $t_0$ is randomly chosen from $\{1, 2, \ldots, T\}$ and independent of $t$. As in the straightening case, additional regularization is necessary to prevent collapse. For a fair comparison, we accomplish this using the same variance and covariance terms used in the straightening model, and thus the total loss is identical to that used in [3],

$$L = L_{\text{invariance}} + \lambda L_{\text{variance}} + \gamma L_{\text{covariance}}. \tag{4}$$

**Synthetic video sequences.** For training data, we generated artificial videos by applying temporally structured augmentations, intended to mimic natural transformations, to static images in common image datasets. The reasons for this choice, as opposed to using a dataset of natural videos, are multifaceted. First, we want to match other image-based SSL models in terms of training data and evaluation pipeline for a direct comparison. Second, models trained on natural videos are known to struggle with image recognition tasks because typical video datasets lack sufficient object class variety [29] (for example, object-centric natural video datasets such as ImageNet VID [30] or Objectron [1] contain only 30 and 9 object classes, respectively). Efforts are being made to align the data distribution of the two domains, but well-accepted benchmarks have not been established yet. Finally, while data augmentation is widely used for generating distinct views of the same image [7], it is uncommon to introduce temporal correlations in the applied transformations (since the goal is to maximize the richness of the training set). This however allows us to create image sequences that have predictable temporal structure that the straightened representation can latch onto.

To create temporally consistent geometric transformations, we can do either of the following: 1) construct a cropping window of a pre-determined size, then gradually move the window in one direction ("translation"), akin to smooth gaze changes; 2) fix the center location of a cropping window, then monotonically increase or decrease its size ("rescaling"), akin to approaching or receding objects; 3) fix the location and size of a cropping window, and rotate the window ("rotation"). We combine these with appearance transformations, in which we linearly adjust the photometric parameters (brightness, contrast, saturation, and hue) across frames within a sequence. This mimics gradual changes in lighting conditions over time. The rate of these geometric and photometric transformations is held constant within each sequence, but varies across sequences.

For the first set of experiments, we created a *sequential MNIST* dataset, in which images of single digits are transformed according to one of the three geometric transformations (Fig. 2A). Each frame contains one digit, randomly selected from the MNIST dataset, moving inside a $64 \times 64$ patch. Each video is 20 frames in duration. For translations, the digits were placed at random locations initially; for rescaling and rotation, the digits are always at the center of the patch. Transformations ramp linearly over time with two exceptions: for translations, digits "bounce" off of the edges, abruptly changing direction, while for rescaling the direction of enlargement or contraction reverses if the size exceeds a preset range. These special cases generate motion discontinuities, where prediction is expected to fail.

For a second set of experiments, we generated a *sequential CIFAR-10* dataset, each with a duration of three frames (Fig. 4A). We eliminated the rotational transformation, as it tends to create boundary artifacts on the nonzero background. Following standard practice in self-supervised learning [7], we also included random horizontal flips, grayscale, and solarization to increase dataset diversity. Horizontal flips, if present, were applied to all frames in the sequence to preserve the frame-to-frame spatial relationships, while the other transformations were applied independently to each frame.

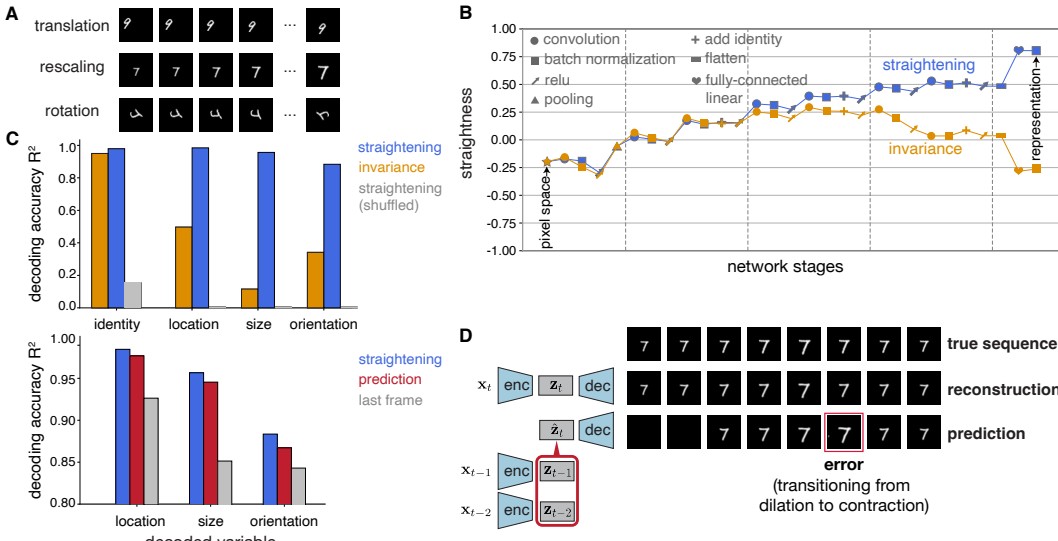

**Figure 2:** Straightening and its benefits, evaluated on a network trained on sequential MNIST. **A**. Three example sequences, illustrating the three geometric transformations. **B**. Emergence of straightness throughout layers of network computation. **C**. Accuracy in decoding various (untrained) variables from the network responses (top). Accuracy in predicting variables at the next time step (bottom). Identity decoding was not considered for prediction as it is constant over the sequence. **D**. Prediction capabilities of the network. Top: example sequence, with dilating/contracting digit. Middle: reconstructions from simultaneous representation. Bottom: predictions (linear extrapolation) based on the representation at the previous two time steps.

**Network architecture.** The network architecture is dataset specific. For sequential MNIST, we instantiated $f_\theta$ as a 7-layer convolutional neural network with simple half-wave rectifier (ReLU) nonlinearities and no skip connections. For sequential CIFAR-10 we used ResNet-18 [20] as the model backbone and attached to the end a projector with 3 fully-connected layers, following the standard practice in [3]. Throughout the experiments we used the same model architecture for the straightening objective (2) and the invariance objective (4); the hyperparameters were optimized separately for each loss so that both models achieve their best recognition performance.

## 3 Straightening learns meaningful representations

How straight can the representations become, and how is straightening achieved? To answer these questions, we trained $f_\theta$ on sequential MNIST and measured the straightness of embeddings at each stage of the network (Fig. 2B). Embeddings learned by the straightening objective (2) are progressively straighter throughout the network, with the largest increases occurring near the last layer (on which the loss function is imposed). In contrast, those learned by the invariance objective (4) initially increase in straightness but then decrease, ending at a value slightly more curved than the pixel-domain input, consistent with observations in [22]. All linear operations (convolutions, spatial blurring, and fully-connected linear projections) contribute to increasing straightness, whereas the rectifying non-linearities usually reduce it. Geometrically, this is because the rectifiers project the embeddings onto the positive orthant, bending temporal trajectories that cross rectifier boundaries.

To better understand the nature of the representations, we visualized the temporal trajectories of digit sequences using a 2D t-SNE embedding [32]. Fig. 1B shows 200 trajectories from the translation subset, in both the pixel domain (left) and the learned representation (right). Even in this non-linear, low-dimensional projection space, the individual representation trajectories are noticeably straighter than their pixel-domain counterparts. Furthermore, the representations clearly separate the digit classes, despite the fact that training was unsupervised, with no explicit knowledge of digit identity.

**Decoding untrained visual features.** Ideally, straightened representations should encode *all* predictable information in the input video and *nothing more*. Explicit predictions can be made in the representation space by linear extrapolations. The design of our dataset ensures that visual features such as location, size, and orientation of the digits are predictable, and therefore, should be preserved in straightened representations. On the contrary, representations learned by the invariance objective should be agnostic to any temporally-varying information, including these features. To test this hypothesis, we trained a support vector machine (SVM) regressor with radial basis kernels (RBF) to read out those attributes from the learned representations. We also trained a linear classifier to decode digit identity. Fig. 2C shows that digit identity can be read out from both models, while only the straightened representations maintain the dynamic features of the inputs. Notice that respecting temporal order is critical for straightening to learn anything meaningful – training the same objective on temporally shuffled frames gives poor decoding performance. This demonstrates another distinction between straightening and invariance: the latter is, by definition, agnostic to temporal ordering.

To test predictability, we used the same decoder to read out location, size, and orientation in the *next frame* from the linearly extrapolated (predicted) representations. We compared this prediction performance against a naive control that simply uses representations of the previous frame. The performance of the straightening predictor is found to be substantially better than the control, and nearly as good as the decodability of the current frame. To further examine the image information contained in the straightened representations, we froze the representations and trained a decoder network (another 7-layer convolutional neural network) to reconstruct frames at the pixel level. We then used the same decoder to visualize the predicted responses given by linear extrapolation. We found that this enables accurate prediction of future frames (Fig. 2D, bottom; additional examples in Appendix), despite the fact that our learning objective did not explicitly optimize for such reconstruction error. Exceptions occur when transformations change direction: switching between expansion/contraction; or bouncing off of boundaries. In particular, the predicted "7" continued to expand when the actual inputs suddenly began to contract. This shows that the representation is able to capture smooth transformations in the input, but not the macro-structure of the transformation statistics.

## 4 The geometry of straight representations

To understand how straightening enables class identification, we analyzed the representation geometry induced by the straightening loss. The t-SNE embedding in Fig. 1B implies that the temporal trajectories of images with the same digit and transformation type are more parallel than expected by chance. To verify this intuition, we computed the cosine similarity of velocity (difference) vectors drawn from trajectories of the same digit and the same transformation (Fig. 3A). This is compared to the equivalent measure for the representation obtained via the invariance objective, and to the distribution expected for random vectors in the output space ($d = 128$).

For comparison, we also measured the cosine similarity of trajectories from different classes. Compared with the random distribution, and the values computed from the invariant representations, the straightened representations showed a substantial bias toward parallelism for trajectories within the same [digit, transformation] class, and toward orthogonality for trajectories across different classes. We hypothesize that trajectories from the same [digit, transformation] class are more likely to have samples that are close to each other in the representation space; these proximal points may have similar local gradients since the model architecture is composed of locally affine operations. When the straightening loss straightens trajectories from the same class, it does so in similar directions, resulting in a larger cosine similarity. On the other hand, the whitening regularizer tries to fill the output space, which encourages orthogonality of trajectories from different classes in order to achieve a high variance in every output dimension. This leads to a cosine similarity for across-class vectors that is more concentrated at zero than expected by chance.

We further corroborated this finding by quantifying the dimensionality of the within-class and across-class network responses. Specifically, we quantified "effective dimensionality" using the participation ratio, $R = \frac{\left(\sum_i \lambda_i\right)^2}{\sum_i \lambda_i^2}$, where $\{\lambda_i\}$ are the eigenvalues of the covariance matrices of responses in each condition. It is clear from Fig. 3F that representations from the same [digit, transformation] class are much more compact under straightening than under invariance, although the

straightened representations exhibit a higher effective dimensionality overall.

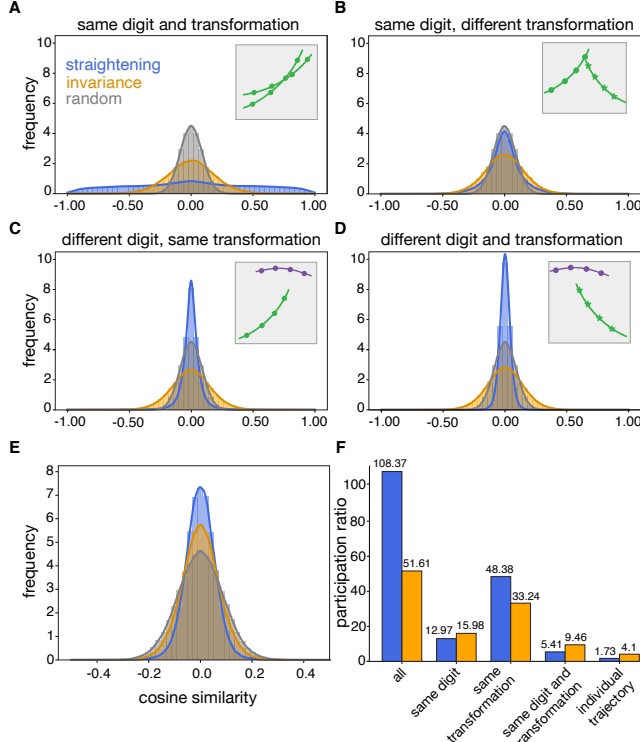

**Figure 3:** Geometric properties of the straightened representation. Panels **A-E** show histograms of cosine similarity (normalized dot product) between pairs of difference vectors, $z_t - z_{t-1}$. Insets show example trajectories in each scenario, where color indicates digit identity. **A**. same digit and transformation type; **B**. same digit and different transformation; **C**. different digit and same transformation; **D**. different digit and transformation; **E**. all difference vectors vs. digit classifier vectors. **F**. Average effective dimensionality, measured with participation ratio, of the set of responses $z_t$ in each group.

How can this geometry give rise to class separation? We argue that if the representations of images in the same class lie in a low-dimensional space, then a classifier can be constructed by projecting out this subspace, and thus eliminating the within-class variation. To test this hypothesis, we computed the cosine similarity of the classifier's decision axis and the trajectory velocities (Fig. 3E). Compared with the distribution of random vectors and the invariance case, the straightened trajectories are more orthogonal to the classifiers, confirming our intuitions. Thus, information about other visual features is preserved in the null space of the decision axis for digit identity.

## 5 Straightening increases recognition robustness

While typical recognition models do not naturally yield straight representations, explicitly training such models for noise robustness can increase straightness as a by-product [31, 19]. Here, we show that the converse is also true: straightening makes recognition models more immune to noise. Unless specified otherwise, for these experiments we focused on the sequential CIFAR-10 data (Fig. 4A) and used ResNet-18 as the backbone representation network. Following the standard practice in [3], we attached to the end of the backbone a projector with the training loss applied to its output. The outputs of the backbone were taken as the primary representations and used for the downstream recognition task. Representations were learned by applying the self-supervised learning objective to clean image sequences. After learning, the network parameters were frozen, and linear classifiers were trained to identify the corresponding image class. Hyperparameters were chosen to yield the best clean image recognition performance, individually for each objective. We used an adapted version of the solo-learn library to train all models in this section [10].

As a variation of the original learning setup, we added a second straightening loss, Eq. (1), to the outputs of the first ResNet block. The two straightening losses were equally weighted and averaged, while the whitening regularizer was only applied at the last layer. We empirically found that using straightening at multiple stages of the processing hierarchy can further improve robustness.

Fig. 4B shows the straightness level of embeddings across every transformation of the network. Similar to the sequential MNIST case, straightness increased sharply near the layer where the straightening loss was applied, and gradually elsewhere under convolution, spatial pooling, and fully-connected linear layers. In contrast, representations optimized for invariance were not straightened across processing stages. Examples of image sequences that produced the straightest trajectories (Fig. 4C, left) show clearly identifiable object contours and temporal transformations, while those

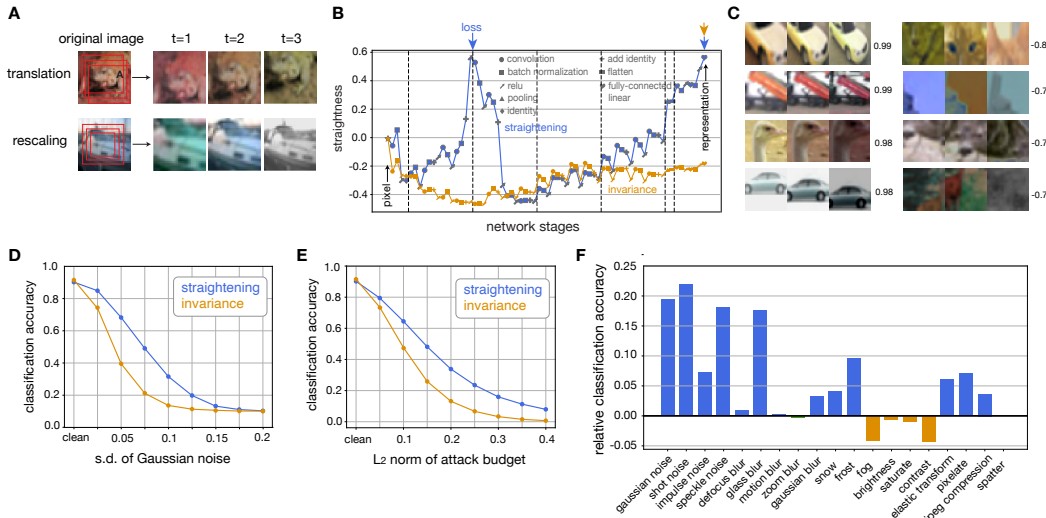

**Figure 4:** Effect of straightening on representational robustness. **A**. Two example synthetic sequences from on sequential CIFAR-10 dataset. Top: translation and color shift. Bottom: rescaling (contraction) and color shift, last frame randomly grayscaled. **B**. Emergence of straightness throughout layers of network computation. Top arrows mark the stages of representation directly targeted for straightening (blue) and invariance (orange). **C**. Example sequences illustrating successes (left) and failures (right) of straightening. Numbers indicate straightness level $\in [-1, 1]$. **D**. Noise robustness: classification accuracy as a function of the amplitude of additive Gaussian noise injected in the input. **E**. Adversarial robustness: classification accuracy as a function of attack budget (see text). **F**. Relative classification accuracy of straightened network compared to invariance-trained network for various degradations. Color indicates the objective with better performance.

that had the most curved trajectories (Fig. 4C, right) appear fuzzy with little identifiable structure and are inherently less predictable.

**Straightening is more robust than invariance.** First, we assessed network robustness by evaluating recognition performance for images with increasingly larger levels of i.i.d. Gaussian noise added to the pixels (Fig. 4D). We found that the straightened representations proved substantially more robust over a wide range of noise levels with negligible degradation in noise-free recognition performance. Second, we tested robustness to adversarial perturbations, which are considered a hallmark failure of artificial vision models [12]. Not only are these models highly susceptible to small amounts of adversarial noise, but the adversarial examples are barely visible to humans, a notable discrepancy between artificial and biological perception. Therefore, adversarial robustness is an important metric of how brain-like the representations are. We used untargeted projected gradient descent (PGD) with the $L_2$ norm constraint to generate adversarial perturbations [13]. For all attack budgets, we chose a step size that is 1/10 of the budget and set the number of PGD steps to be 500 to ensure that the attack optimization procedure had fully converged. We found that the straightening objective substantially increased the robustness to white box attacks over all attack budgets without degrading performance on clean images, as shown in Fig. 4E.

We also tested recognition robustness on a composite of corruptions. We used the CIFAR-C dataset [24], which defines 18 types of corruptions coarsely grouped into "noise", "blur", "weather" and "digital" categories and evaluated the models on corruptions of the highest intensity. Fig. 4F shows the relative performance of straightening versus invariance. Straightened representations were again more robust than those optimized for invariance for many forms of image corruption, most notably those in the noise category. All corruption types for which invariance proved superior to straightening were directly (brightness, saturate, and contrast) or indirectly (fog, as a specific form of contrast degradation) included as augmentations in the training set, and thus their robustness was a natural consequence of the invariance objective. Overall, these results demonstrate that learning by straightening brings with it a systematic benefit in robustness to a wide variety of input

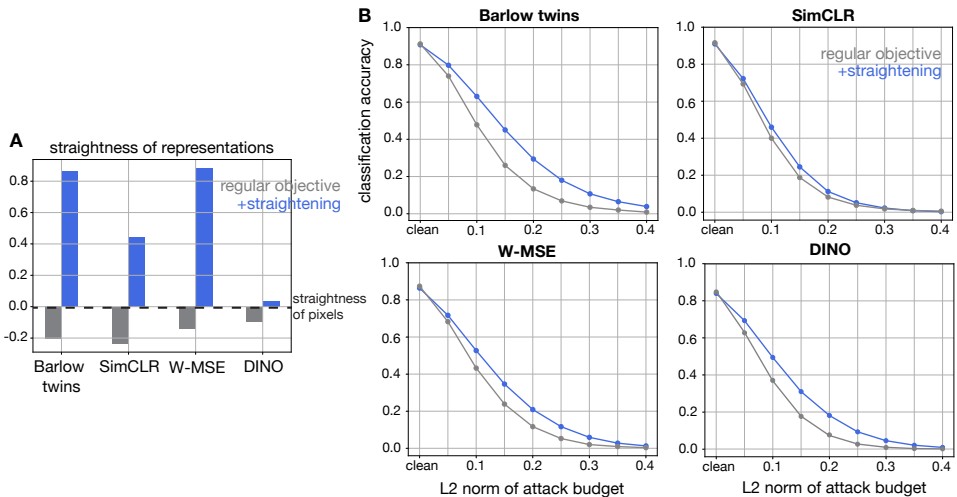

**Figure 5:** Augmentation of other SSL objectives with a straightening regularizer. **A**. Straightness of representations learned by four different SSL objectives (gray), and their augmentation with a straightening regularizer (blue). **B**. CIFAR-10 classification accuracy as a function of adversarial attack budget, for the original and straightening-regularized version, for the same four SSL objectives.

degradations, without the computational costs and complexity associated with directly optimizing for such robustness.

Straightening also improves representational robustness for natural temporal transformations. Specifically, we used the same training procedure on the EgoGesture video dataset [5, 36] for egocentric hand gesture recognition. For simplicity, we used the depth channel as input, which circumvents the need for modeling the background. To preserve the motion structure we did not apply any frame augmentations. For this dataset, we found that the straightened representations were more robust than invariance-based solutions across a wide range of corruptions. See Appendix for results and implementation details.

**Straightening improves robustness in other SSL models.** In principle, straightening can be incorporated into any SSL learning objectives as long as the inputs are a temporal sequence of at least three frames. Can straightening robustify representations when combined with other SSL losses? To answer this question, we regularized several existing SSL objectives using our straightening loss (1), trained the models on sequential CIFAR-10, and tested their recognition performance under adversarial attacks. In all cases, the straightening loss was added to the outputs of the projector as well as the first ResNet block, with the weight chosen from a parameter sweep that optimizes recognition performance. Other hyperparameters of the SSL models, including weights of other terms in the objective, and the architecture of the projector, were kept to their original values/setup.

None of the original SSL models demonstrate straightening in their representations (Fig. 5A, gray), but modest amounts of straightening regularization can significantly improve straightness beyond the pixel level (blue). Fig. 5B compares adversarial robustness under the original SSL loss [35, 7, 14, 6] and the corresponding variant regularized by straightening. For all objectives tested, straightening systematically improved representational robustness, even though the original training was already heavily tuned to optimize performance. We repeated the same robustness test on pixel-level white noise (see Appendix) and observed the same benefits of adding the straightening loss. This suggests that the idea of representational straightening and the use of temporally smooth image augmentations may prove of general practical utility for robust recognition, and makes straightening an important new tool in the SSL toolkit.

## 6 Discussion

We have shown that a biologically-inspired SSL objective that promotes straightening in the representation of image sequences leads to predictive neural representations that factorize geometric,

photometric, and semantic attributes of the input. These embeddings also prove more robust to various forms of noise and other degradations, compared to SSL methods that optimize for augmentation invariance. Moreover, incorporating straightening as a regularizer extends these benefits to other SSL training procedures, suggesting a broader utility for straightening as a cost-effective mechanism for robust unsupervised learning.

Directly improving robustness to adversarial attacks via optimization is difficult and computationally costly [12]. In contrast, our solution achieves similar results in an easy and computationally less demanding manner. The key ingredient for its success is having meaningful temporal structure in the input sequences. This could either come naturally through the use of video (although good datasets for that are scarce [29]) or, more practically, can be artificially enforced by temporally correlated image augmentations. Thus, our results highlight a new useful form of data augmentation in support of learning predictive representations.

In contrast to invariance, which aims to map all elements of a semantic class into unique points in representational space, discarding all within-category variability, straightening strives to encode all structured across-frame variations in the input. In doing so, it produces rich image embeddings containing structured information about class identity, as well as various transformations which are represented in different subspaces, and thus easy to decode. Geometrically, this means that straightening leads to overall higher dimensional embedding spaces but the individual semantic components (image class, or class $\times$ transformation) are much lower dimensional. This joint increase in embedding dimension and reduction in manifold dimensionality increases the model's representational capacity [9, 33], which may explain its increased robustness.

We have advocated for the replacement of hand-selected augmentations with the readily available temporal structure of natural visual experience [37, 29]. However, the predictable structure of natural videos evolves at multiple timescales. It is not clear whether a feedforward architecture that operates on one frame at a time and makes predictions at a single temporal scale is enough to fully exploit such structure. As the predictable horizon of different elements in our visual input varies across levels of abstraction, a natural extension would be to enforce straightening at multiple time scales and in multiple network stages. More work is needed to determine how to best incorporate a hierarchical temporal structure in the straightening loss to accommodate long horizon predictions, but we expect this type of hierarchical prediction to play a central role in developing models for both biological and machine vision.

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

# Appendix

## A   Pretraining details for sequential MNIST

To find the optimal weights in the loss function (2), we used a parameter sweep to choose the $(\alpha, \beta)$ pair that gives the best clean image recognition performance, with resulting choices $\alpha = 1.0$, $\beta = 0.25$. For the invariance objective, a similar parameter search yields $\lambda = 0.125$, $\gamma = 0.5$. The detailed architecture of the encoder is described by the sequence of transformations in Fig. 2B.

## B   Pretraining details for sequential CIFAR-10

For data augmentation, each frame has an independent probability to be grayscaled ($p = 0.1$) or solarized ($p = 0.1$). Each sequence has a probability of $p = 0.5$ to be horizontally flipped. For the straightening objective we set $\alpha = \frac{15}{9}$ and $\beta = \frac{1}{9}$. For the invariance objective, we used the reported weight parameters from [3]. When straightening was added to the main SSL objective as a regularizer, we kept the default optimal weight parameters taken from solo-learn [10] in the main objective and only tuned the weight of the straightening loss. We set this weight to 3 for barlow twins, 0.1 for SimCLR, 0.2 for W-MSE, 0.005 for DINO.

We used the LARS [34] optimizer with learning rate $0.3$, weight decay $1e - 4$, batch size 256 to train our straightening model. For all other models we used the default setting given in the solo-learn library. All pretraining was run for 1000 epochs, which takes roughly 5 hours on 4 A100 Nvidia GPUs.

## C   Additional pixel-level reconstruction and prediction results

Here we provide more reconstruction and prediction examples for translation, rotation, and rescaling. The full sequence is shown here ($t = 20$) in the same format as in Fig. 2D.

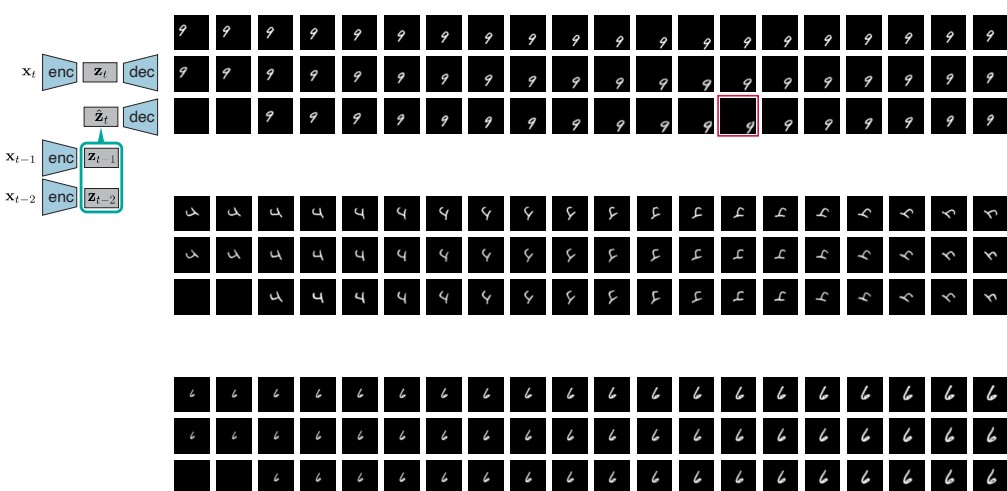

## D   Straightening robustifies other SSL models on white noise

Here we additionally show CIFAR-10 classification accuracy as a function of the intensity of white noise added to the pixel input. Adding the straightening regularization to the main objective function improves recognition robustness under white noise for all four existing SSL models we tested.

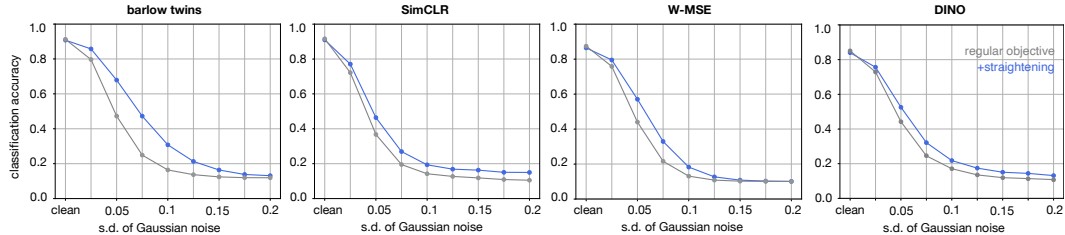

## E    Straightening natural temporal transformations

**Pretraining details**    We used ResNet-18 as the backbone architecture together with a projector with 3 fully-connected layers, but modified the first convolution to accept the single channel input. We applied either the straightening objective (2) or the invariance objective (4) to the outputs of the projector. The models are trained from scratch with no pretrained weights. No data augmentation is applied, so models can only rely on the natural frame-to-frame variations. We choose 6 frames with a fixed interval ($4\Delta t$) from each gesture clip. While this might not be the optimal setting if the goal is to optimize performance on gesture recognition, our purpose is to compare the straightening and the invariance learning objective in exactly the same setup.

**Evaluation pipeline**    For gesture classification we freeze the model and concatenate the outputs of the backbone of all 6 frames to train the linear classifier. To test robustness we add Gaussian noise of various levels to pixels. The straightened representations are more noise robust than the invariance-based solutions.

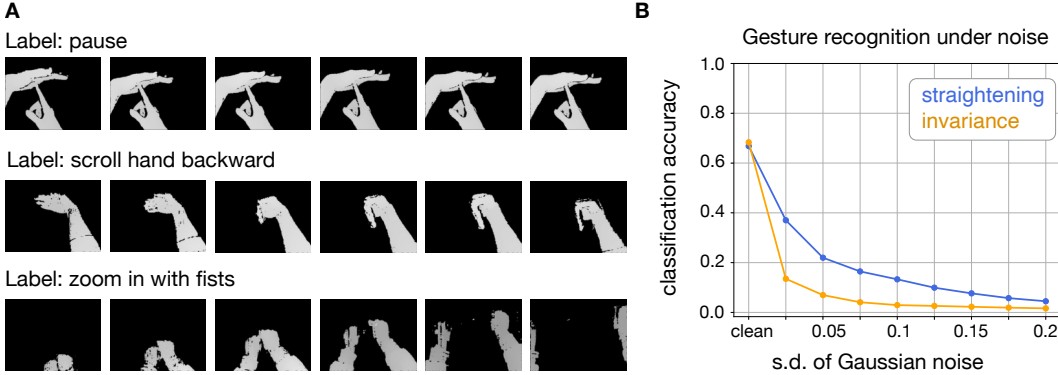

**Figure 6: A**. Example gestures. Some gestures can be classified by a single frame (pause), while others must observe multiple frames to recognize the motion (scroll hand backward, zoom in with fists). **B**. Gesture recognition performance as a function of noise level.

