# OpenReview forum: "Learning predictable and robust neural representations by straightening image sequences"
_NeurIPS.cc/2024/Conference — NeurIPS 2024 poster_

### Official Review · Reviewer_Z5Vh · 2024-06-28

**Soundness:** 4
**Presentation:** 4
**Contribution:** 2
**Rating:** 7
**Confidence:** 4

**Summary:**

This paper presents a simple self-supervised learning objective which aims to "straighten" representation trajectories in latent space - maximizing cosine similarity of consecutive deltas of representation (i.e take three time steps, calculate the difference in representation between each pair of consecutive representations and calculate the cosine similarity over these differences).
To prevent collapse the author suggest using two common regularization losses - one pushes the variance of each representation dimension to one and one decorrelates the representation dimensions.
The objective is used to train networks on two simple, synthetic sequential datasets and is demonstrated to learn interesting representations demonstrated in a variety of ways - readout accuracy, robustness to noise and adversarial attacks.

**Strengths:**

*Originality:*
While based on some existing work (and very close to [14], as the authors note) I found the work original in the way it is applied and actually simpler than existing work (a good thing!). I also appreciate the general context of the work and the neuroscience connections that can be made here,

*Quality:*
This is very solid work - while the idea is simple, the analysis and breadth of the _existing_ experiments are very good. I appreciated the multiple angles the learned representation is inspected - statistically across the data, depth wise across layers and downstream task readouts. This is a nice example of how investigating the results of a method shed light on the "idea" of a paper and its significance.
Having said that, see below for some comments on the scope of experiments at large.

*Clarity:*
The paper was a pleasure to read, with clear and concise language, no attempt to make things more complicated than they are and good figures and captions. See below for some (minor) comments.

*Significance:*
Probably the weakest point of the paper - this is a very "small" paper in scope and (see reasons below) I think its actual contribution to the community is a bit limited. Having said that, it is an "idea" paper (rather than a "results" paper) so could serve as a basis for future work increasing the potential contribution.

**Weaknesses:**

*Scope of experiments:*
While I appreciated the depth and breadth of the existing experiments I think the scope of the experimental validation is still a bit limited. I appreciate the difficulty in running larger scale experiments with video data for many groups due to resource limitations, however, I feel that in this case, the use of synthetic data alone, and beyond that, just MNIST and CIFAR10, is a very limiting factor. I would have loved to see experiments on larger, natural(istic) datasets. For an objective like the one proposed in the paper, using sequences with very simple, almost linear transformations across time makes the results a lot less interesting than they could have been - no wonder that a model trained to learn simple transformations actually does it with data which is only comprised of simple transformation. The interesting question is what this model can do when the transformation are complex and non-linear as the ones in natural video - multiple objects moving, viewpoint changes etc.

Beyond that I would be interested to see measures of straightness to some existing, pre-trained models - especially from the family of state-of-the-art SSLs like DINO v2, MAE etc. Do they naturally learn "straight" representation even though not trained on sequences? can you fine-tune them with a straightening loss?

To be clear, I do not expect the authors to run any new experiments during the rebuttal period as I appreciate what I am suggesting here may take more time and resources than is possible for this short amount of time (but needless to say I'm happy to see any results the authors may be able to produce in this context).

*Presentation (minor):*
While I enjoyed the presentation in general I think the use of coloured labels (Figure 2C for example) right on the plot is weird and confusing. A legend would have been more useful. A bit more content in the captions would have been nice as well - Figure 3, for example has insets which are not explained in the caption at all.

**Questions:**

Beyond the comments above I have some more minor questions:

1. The text mentions that in order to calculate the "straightness" the representation needs to be flattened - while this makes sense, I do not understand why we see such a big jump in "straightness" around flatenning layers (Fig 2B and Fig 4B) - I would expect this layer to have no effect (because the flattening happens either way) but it's the biggest "jump" in the plots.

2. The authors suggest that adversarial examples for common networks have imperceptible perturbations and it is implied, I think, that adversarial examples for straightening models are different - but I think these are not shown in the paper, am I wrong?

**Limitations:**

The authors address some of the limitations in the paper, I hope to see a bit more of them in light of what's written above.

---

> ### Author Rebuttal · Authors · 2024-08-07
>
> Thank you for your review and comments.
>
> **Scope of experiments:** We acknowledge the limitations of our current experiments. In the global response, we explained why we did not use natural videos, and our plan for improvement.
>
> **DINO v2 and MAE:**
> - To give a partial answer to the reviewer’s questions, we tested pre-trained DINO on our sequential CIFAR dataset (code adapted from [8]). The representations are not naturally straightened. In fact, none of the SSL models we tested in Figure 5 naturally shows straightening; representations become straighter only after the straightening loss is added to their original loss. For DINO, adversarial robustness also improves after augmented with straightening. We provided Figure R3 in the rebuttal PDF to emphasize this result.
> - In the short time frame we were not able to find a reliable implementation of DINO v2 or MAE trained on CIFAR10. But it was shown in [16] that MAE (see Figure 24) and ViT-based DINO (see Figure 7) do not naturally straighten natural videos.
>
> **Presentation:**
> - In the rebuttal PDF we provide an updated version of Figure 2C with legends, where we also added regression results for the shuffled case.
> - The insets of Figure 3 show a schematic diagram of the representation geometry for each case. A) Trajectories are more parallel if they are from the same digit and the same transformation class; B) C) D) Trajectories are more orthogonal if they are from different classes. We will revise the caption to clarify this.
>
> **Flattening layer effects:** Flattening has no effect on straightness – in fact, they are included as a sanity check. But spatial pooling and fully-connected linear layer can improve straightness because by selectively projecting responses to a lower dimension, they alleviate noise and represent coarser and smoother features [19].
>
> **Qualitative assessment for adversarial images:** We did not make a claim regarding the adversarial examples for straightening modelsr - see additional discussion in the global response.
>
> **Expanding the writing for the limitation section:** We will expand the discussion of the scope of the work in the Limitations section of the updated paper.

---

> > ### Comment · Reviewer_Z5Vh · 2024-08-12
> > **Thank you for the detailed response.**
> >
> > I thank the reviewers for the time and effort taken to respond to my (and other's) review.
> > As I mentioned in the original review I think that is very interesting work that should be accepted - many of my concerns have been answered in the rebuttal and the ones that haven't been (I still think natural video would have been cool to see here) are not a reason to reject the paper.
> > I am therefore increasing the score and am looking forward to see this line of work evolve in the future.

---

### Official Review · Reviewer_jxcp · 2024-07-11

**Soundness:** 2
**Presentation:** 3
**Contribution:** 3
**Rating:** 6
**Confidence:** 4

**Summary:**

the current manuscript introduces a self-supervised learning (SSL) objective inspired by biological vision systems. It proposes an objective that promotes the "straightening" of neural representations of image sequences, facilitating linear prediction. The proposed method is tested on small and synthetic datasets like sequential MNIST and CIFAR-10, demonstrating that the learned representations are more predictive, robust to noise and adversarial attacks, and can enhance other SSL methods when used as a regularizer.

**Strengths:**

- The paper is relatively easy to follow and well structured.
- The paper is motivated by findings from biological vision systems, providing a novel angle to the objective function design in SSL.
-  The paper provides a clear geometric explanation of how *straightening* contributes to class separability, which helps in understanding the underlying mechanics of the proposed method.

**Weaknesses:**

- The main weakness is the novelty claim regarding the objective function. Similar loss functions and concepts have been explored in other works. Examples are: [1-5]
- The paper needs to better differentiate its approach from existing methods that use linear predictors or phase-pooling for straightening.
- another major issue is the experimental settings that are primarily conducted on synthetic datasets. While this allows controlled comparisons, it limits the demonstration of the method's applicability to real-world data.
- and comparisons to existing state-of-the-art SSL methods are somewhat limited. More extensive benchmarking against a broader range of methods could strengthen the claims.

References:
[1] VICReg: Variance-Invariance-Covariance Regularization for Self-Supervised Learning published at ICLR 2022
[2] Learning a Depth Covariance Function published at CVPR 2023
[3] MLVICX: Multi-Level Variance-Covariance Exploration for Chest X-ray Self-Supervised Representation Learning
[4] Variance Covariance Regularization Enforces Pairwise Independence in Self-Supervised Representations
[5] An Information-Theoretic Perspective on Variance-Invariance-Covariance Regularization

**Questions:**

Please see the weaknesses and limitations.

**Limitations:**

The paper presents an interesting approach to SSL by leveraging a biologically inspired objective that promotes the *straightening of neural representations over time*. This approach is evaluated on synthetic datasets, showing improved robustness and predictive performance. However, the novelty of the objective function is not fully established, as similar concepts have been previously explored. Additionally, the reliance on synthetic datasets limits the demonstration of the method's real-world applicability. While the paper provides a clear geometric intuition for its approach and shows promising results, more extensive comparisons with state-of-the-art methods and testing on real-world datasets would strengthen the claims. Therefore, based on the current presentation and evaluation, the novelty and practical impact of the proposed objective function is not sufficiently justified.

---

> ### Author Rebuttal · Authors · 2024-08-07
>
> Thank you for your review and comments.
>
> **General comments on novelty:** see global response.
>
> **How straightening differs from the references mentioned by the reviewer:** References pointed out by the reviewer seem to focus on why and when the variance-covariance regularizer is useful, while our contribution focuses on the straightening objective. In fact, what we showed in Figure 5 is that straightening can be accompanied by various forms of regularization and still learn useful and robust representations. VICReg is one of the main reference models we compared to (and beat!).
>
> **How straightening differs from linear prediction and phase pooling:**
> - Straightening is the simplest form of second-order linear prediction in that it requires no parameter setting in the prediction step. Yet, it demonstrates excellent predictive power for the transformations we tested on (see Figure 2D and Appendix B), and yields unexpected robustness that it is not explicitly trained for. It would not be surprising that a general parametrization of a second-order linear predictor could perform at least as well as straightening, but we would argue that achieving comparable results with a simpler objective corresponds to a significant contribution (rather than  a lack of novelty).
> - [14] uses a complex architecture including things like phase-pooling and further relies on an autoencoder structure and a pixel-level prediction loss to prevent information collapse. Our solution succeeds with a much simpler architecture. Critically, [14] did not show any quantitative evaluation of the learned representations, while we compared our results to state-of-the-art SSL models that are difficult to beat.
>
> **Reasons for not having used natural videos:** see global response.
>
> **More comparison to benchmark results:** While it is unrealistic to compare to all models on the market, we have added a new comparison with the DINO method shown in Figure R3 of the rebuttal PDF. We were able to show that in this instance as well, representations become more adversarially robust when the DINO objective is augmented with a small amount of straightening (weight for straightening is 0.005).

---

> > ### Comment · Reviewer_jxcp · 2024-08-13
> > **response by Reviewer  jxcp**
> >
> > I have carefully reviewed the feedback from other reviewers, considered the author’s rebuttal, and global responses and followed the ensuing discussion. I appreciate the authors' thorough responses, particularly their clarification on W1 and new experimental results (B and C) during the rebuttal period. therefore I will raise my score from 4 to 6.

---

### Official Review · Reviewer_etrK · 2024-07-23

**Soundness:** 3
**Presentation:** 4
**Contribution:** 3
**Rating:** 7
**Confidence:** 4

**Summary:**

This is a very interesting paper that wants to show that robustness is a consequence of perceptual straightening during training -- both areas that have largely remained disconnected in vision. In particular because adversarial robustness is generally studied from a theoretical perspective, or empirical cat-and-mouse scenarios of new attacks vs defense (or privacy). In the case of perceptual straightening, since it has only recently been introduced as a neuroAI-like inspired motif that should be added in machine vision systems it is not clear if straightening will provide robustness or the other way around (determining the causal factor -- though some previous work has shown this to certain degree). This paper shows that straightening a model will provide robustness.

**Strengths:**

* The topic of perceptual straightening is relevant and heavily under-discussed at the intersection of computational vision and representational learning.
* I think it is nice to use an organic version of SSL via temporal conditioning as a way to do perceptual straightening
* The goal of the paper is easy to understand: apply perceptual straightening on a model, then test to see if it will be more robust than its non-straightened version (it is!)

**Weaknesses:**

- The paper could do a better job exploring the  **qualitative assessment** of the response of different training networks (with and with-out straightening and SSL) **for adversarial images**. Authors make a strong case that robustness to adversarial images is stronger for perceptually straightened (PS) neural networks, only showing curves, but is this actually the case if we qualitatively look at samples? Will attacking a PS-NN with a target fish label actually morph the image into a fish? See for example:  Berrios & Deza. SVRHM 2022. Santurkar et al. NeurIPS 2019 and recently Gaziv et al. NeurIPS 2023.

-	In Figure 2C I like that there is a shuffled condition to break the linearity of the straightening. What’s not clear to me is if in the other bar plots within figure C the shuffle condition has achieved 0 decording since it is not visible (eg. location, size, orientation)? Or was the experiment not ran? Would authors be able to clarify this?

**Questions:**

* There is something odd about using synthesized data (of MNIST?) as a proxy for real video data with natural scene statistics. I wish there was an experiment with real video data such as down-sampled video data from YouTube or Autonomous driving video data (real vs shuffled). Perhaps I missed this in the paper, and am curious to know authors thoughts on this.

**Limitations:**

Missing papers:

- Santurkar et al. NeurIPS 2019. Image Synthesis with a Single (Robust) Classifier
- **Kong & Norcia. SVRHM 2021. Are models trained on temporally-continuous data streams more adversarially robust?**
- Berrios & Deza. SVRHM 2022. Joint rotational invariance and adversarial training of a dual-stream Transformer yields state of the art Brain-Score for Area V4
- Gaziv et al. NeurIPS 2023. Robustified ANNs Reveal Wormholes Between Human Category Percepts

---

> ### Author Rebuttal · Authors · 2024-08-07
>
> Thank you for your review and comments.
>
> **Qualitative assessment for adversarial images:** see global response.
>
> **Regression results:** The purpose of training straightening on shuffled frames is to validate that straightening indeed makes use of the temporal correlations of inputs, and that if the correlations are destroyed, learning fails. We did not run regression on location/size/orientation for the shuffled case, because we thought the failure in decoding object identity would be sufficient. But for clarification and completeness, we’ve now verified that decoding accuracies are essentially zero - location/size/orientation information is completely lost in the shuffled case. These results are shown in Figure R2 of the rebuttal PDF.
>
> **Reasons for not having used natural videos:** see global response.
>
> **The Kong & Norcia paper:** Thanks for pointing out this relevant paper, which shows models trained on temporally-continuous data (the SATCam video frames) are more adversarially robust than those trained on ImageNet. We will certainly add a citation. This is complementary to our contribution suggesting that natural video statistics have temporal structure exploitable by contrastive learning. Note however that the straightening objective is fundamentally different from the two objectives used in this paper, namely temporal classification (classifying frames to the episode they belong to) and temporal contrastive learning (frames are positive examples when they are close in time).

---

### Author Rebuttal · Authors · 2024-08-07

We thank the reviewers for their comments and questions. This global response addresses questions that were raised by multiple reviewers, and the other points are addressed in individual responses.

**Why didn’t we use a natural video dataset?**
- To link robustness and straightening, our primary comparisons are to SSL models trained and tested on static image datasets, so we sought to match these in terms of training data and evaluation pipeline.  Under these conditions, our model is on par or better than the competition.
- We had considered the possibility of training both the reference and straightening models on a video dataset. But typical video datasets lack sufficient object class variety [25] (for example, object-centric natural video datasets such as YouTube-BoundingBoxes [Real2017] or Objectron [Ahmadyan2021] contain only 23 and 9 object classes, respectively). Some efforts have started to align the data distribution of the two domains, but well-accepted benchmarks have not been established yet. However, given the shared concern from the reviewers we will include a minimal version of this experiment in the final paper. Under the same training and testing pipeline, we will compare the relative performance of reference models and the straightening model on robustness and object detection.
- In addition to their insufficient object class variety, the predictable structure of natural videos evolves at multiple timescales. It is not clear whether a feedforward architecture that takes in one frame at a time and makes predictions at a single temporal scale is enough to fully take advantage of such structure. As described in the Discussion section, we are currently working on a hierarchical extension of straightening that is better suited to capture the interaction across multiple spatial and temporal scales. We think this will be a separate contribution and is not the focus of this submission. Still, it is worth noting that this additional complexity is not needed for the single-scale straightening model to be on par with or better than SSL state-of-the-art.

**Qualitative assessment for adversarial images**
- Supervised and self-supervised representations exhibit similar performance for generalization and noise robustness [Geirhos2020],  and both are susceptible to imperceptible adversarial attacks.
- For the invariance-based SSL models and straightening: In Figure R1 of the rebuttal PDF, we show two examples of untargeted attacks under a span of attack budgets. For small budgets, their adversarial images are indistinguishable from their original counterparts. When the budget is large (L2 norm above 2.0), attacks generated from the straightening model appear more visually apparent than the invariance counterparts. The straightening attacks seem to concentrate on key parts of the object, while the invariance attacks are distributed throughout the image. This suggests some degree of alignment between the straightened representations and human perception. However, this effect is not easily quantifiable, therefore, we did not include these in the paper. The adversarial images generated for untargeted and targeted attacks do not look qualitatively different.
- More generally, we do not think it fair to compare adversarial robustness performance or adversarial images of robustly trained networks and SSL networks, as the former are specifically trained to correct for the mistakes of adversarial images. Importantly, robustified networks are extremely costly to train while our solution achieves robustness with minimal compute costs.

**Some general comments on novelty and significance**
- Our work builds on a foundation established in several previous publications: 1) straightening provides a particular form of predictive coding, and provides a specific objective that has been shown consistent with biological representations [3, 9, 20]; 2) recent SSL developments provide effective methods for preventing representational collapse [2]; 3) recent empirical attempts to characterize,  post-hoc,  straightening in trained neural networks [16, 26].  The work that comes closest to ours in goals is [14], which tried using straightening as part of a learning objective with a more complex architecture. We are the first to show that straightening can actually achieve competitive SSL performance - specifically: 1) straightening learns richer semantic representations than state-of-the-art contrastive methods (as it keeps more information about the input than invariance objectives) and 2) these representations automatically inherit noise robustness. This makes our form of straightening an important new tool in the SSL toolkit, one which we expect to generalize to other learning setups (at the very least as a useful regularizer). Thus, this paper is just a first step, and we expect the community will build upon and expand our results.

Additional references: \
[Real2017] Real, E., Shlens, J., Mazzocchi, S., Pan, X., & Vanhoucke, V. (2017). Youtube-boundingboxes: A large high-precision human-annotated data set for object detection in video. In proceedings of the IEEE Conference on Computer Vision and Pattern Recognition (pp. 5296-5305). \
[Ahmadyan2021] Ahmadyan, A., Zhang, L., Ablavatski, A., Wei, J., & Grundmann, M. (2021). Objectron: A large scale dataset of object-centric videos in the wild with pose annotations. In Proceedings of the IEEE/CVF conference on computer vision and pattern recognition (pp. 7822-7831). \
[Geirhos2020] Geirhos, R., Narayanappa, K., Mitzkus, B., Bethge, M., Wichmann, F. A., & Brendel, W. (2020). On the surprising similarities between supervised and self-supervised models. arXiv preprint arXiv:2010.08377.

---

### Decision · Program_Chairs · 2024-09-25

**Decision:**

Accept (poster)

**Comment:**

There has been significant interest in the idea of predictive coding, both in neuroscience and also in computer science. Here the authors argue that “straightening” the dynamics of information representation can facilitate predictions and implement this idea in a novel algorithm.

There was enthusiasm among the reviewers for this work, noting that the problem is under-studied in the field, the paper is well written, and praising its simplicity.

Some references were provided by reviewers (especially jxcp) for potential additional benchmarking/comparisons.

The authors provided reasonable and adequate answers to the main concerns.

I share the reviewers’ enthusiasm for this paper.